# Wire-Feeding Laser Welding of Copper/Stainless Steel Using Different Filler Metals

**DOI:** 10.3390/ma14092122

**Published:** 2021-04-22

**Authors:** Xiaoyan Gu, Ziwei Cui, Xiaopeng Gu, Jiaxu Shao

**Affiliations:** Key Laboratory of Automobile Materials, School of Materials Science and Engineering, Jilin University, 2699 Qianjin St., Changchun 130025, China; gxy@jlu.edu.cn (X.G.); czw13029036769@163.com (Z.C.); shaojx1618@mails.jlu.edu.cn (J.S.)

**Keywords:** wire-feeding laser welding, copper, stainless steel, digital image correlation, electron backscatter diffraction

## Abstract

Ni-based filler metal and Ni-Cu-based filler metal were used to obtain copper/stainless steel (Cu/SS) joints through wire-feeding laser welding. Along the SS/weld interface, there exist different grain sizes (from coarse columnar grains to fine equiaxed grains). The heat affected zone (HAZ) on the copper side consisted of two areas with different grain sizes and the size of the grain in the Cu-HAZ of the Ni-Cu-based filled joint was much smaller than that of the Ni-based filled joint. Our results showed that grain refinement at the copper/weld (Cu/weld) interface of the Ni-Cu-based filled joint was observed through high-resolution electron backscattered diffraction (EBSD). There was a hardness elevation at the Cu/weld interface of the Ni-Cu-based filled joint due to the grain refinement on the weld of the copper side. The maximum tensile strength of the Ni-Cu-based filled joint was obtained and reached 91.2% of the tensile strength of the copper base metal (Cu-BM). Joints in this study were observed to fracture in a ductile mode. Furthermore, the Ni-Cu-based filled joint exhibited a higher plastic deformation, which was primarily caused by the large deformation of the weld zone and the large deformation of the Cu-BM due to the high plasticity of the weld, which alleviated the stress concentration, as indicated by 2D-digital image correlation (DIC) test results.

## 1. Introduction

Because of the excellent thermal conductivity and ductility of copper-based alloys, and the excellent resistance to corrosion of austenitic stainless steel, both have been widely used in the power generation industry. Copper/stainless steel (Cu/SS) joints, not only have relatively high electric, heat characteristic, and corrosion resistance, but are also light in weight and have a high strength. Nowadays, Cu/SS joints have been widely used in the nuclear, petrochemical, and chemical industries. Although no brittle intermetallic compounds are formed between Cu/SS, the physical and chemical properties of Cu and SS are quite different (especially melting point and thermal conductivity), resulting in metallurgical challenges with Cu/SS joints [1]. Previous research showed that during welding, poor welds could be formed due to rapid heat loss, which was caused by the large thermal conductivity of copper [2,3].

Various fusion welding methods have been applied to overcome the welding problems mentioned above. Previous work mainly focused on two aspects. On the one hand, some studies focused on changing the offset of the laser beam or electron beam to obtain defect-free joints. Meng [4] showed that defect-free joints could be obtained when the laser offset was 0.5~1 mm on the copper side; joints under the optical laser offset all fractured at the copper side and the achieved tensile strength was 90% of the base copper metal, and when the laser offset exceeded 1 mm, coarse copper columnar grains were formed in the weld, which deteriorated the mechanical properties of the joints. Guo [5] showed that when the electron beam offset was −0.2 mm to 1 mm, the mechanical properties of joints was excellent due to the narrow HAZ and defect-free weld zone. When the offset was not in the range of the optical offsets, welding defects such as the deterioration of the HAZ were found. Kuryntsev [6] achieved a defect-free copper/steel fiber laser welded joint by focusing the laser point on the steel side; the stainless steel was melted by the laser while the copper was melted by the conducted heat, and the Cu-HAZ consisted of two regions with different grain sizes. On the other hand, some studies produced a sound laser welding (Cu/SS) joint by effectively controlling the melting of Cu in the welding fusion zone [7,8,9]. Previous studies found that the copper content in a weld zone was significant to the enhancement of joint performance. Studies conducted by another group found that defects can be reduced by reducing the copper content in a weld. However, composition inhomogeneity and severe microcracks in the weld zone would be formed by the higher amount of copper.

Moreover, the methods of changing the welding parameters put forward high requirements on welding accuracy.

Vacuum conditions are harsh in electron beam welding [10,11]. As laser welding works well under normal conditions and has a high energy density, small post-welding deformation, and high welding efficiency, it has been widely applied in practical engineering joining. As the major drawback in Cu/SS laser welding is stringent joint tolerances, the widest acceptable gap is usually 10% of the material thickness in the butt joint, and this condition is often difficult to reach [12]. Because of convenient metallurgical adjustments and unlimited joint forms, wire-fed laser welding has no restrictions on the joint form [13,14,15], which possesses great advantages in improving the welding accuracy of thin plate welding for dissimilar materials, and has been widely used in the connection between Cu/SS composite components. However, due to the laser heat input and compositional gradients in the weld zone, local solidification processes, as well as grain growth, have changed. Although a defect-free joint can be obtained after welding, there was still solute segregation in the weld zone and grain-coarsening in the Cu-HAZ [16], which affected the integrity of the structure in the weld zone and reduced the strength of the welded joint, respectively. In previous studies, research on the microstructure of wire-fed laser welded joints and the correlation with mechanical properties was not in-depth.

Consequently, wire-fed laser welding was used to obtain Cu/SS dissimilar metal joints in this study. Ni-based and Ni-Cu-based filler metals were used to weld Cu/SS dissimilar metals, the microstructure in Cu-HAZ and the unmelted zone (UZ), the solidification mode and the grain refinement in the weld zone were analyzed to further enrich welding theory. EBSD was used to analyze the grain refinement at the Cu/weld interface of a Ni-Cu-based filled joint. Additionally, a 2D-digital image correlation (DIC) test was conducted to reveal the local microstructure–mechanical property relationships.

## 2. Materials and Experimental Methods

### 2.1. Materials

In this study, the base metals (BM) were T2 copper (Cu) and 301 austenitic stainless steel (301 SS) sheets. The Cu-BM and 301 SS-BM (thickness of 1 mm) were machined to rectangles with sizes of 100 mm × 50 mm. The Ni-based filler was 600 alloy and Ni-Cu-based filler was ERNiCu-7, respectively, the diameters of which were 1.2 mm. According to the Standard of Specification for Annealed or Cold-Worked Austenitic Stainless-Steel Sheet Strip, Plate, and Flat Bar and Standard Specification for Copper Sheet, Strip, Plate, and Rolled Bar, the chemical compositions of the base metals are listed in Table 1. The chemical compositions of the welding wires provided by the supplier are listed in Table 2.

### 2.2. Welding Process

The welding machine was an RC-YLS-2000 continuous fiber laser (IPG, New York, NY, USA) with a rated power of 2 kW and the diameter of the laser beam was 0.2 mm. In order to remove the oxide film, sandpaper was used to the polish samples before the welding. Welding tests were carried out on the butt joint and 99.99% pure argon gas (with a flow rate of 10 L/min) was adapted to protect the weld bead. As shown in Figure 1, the constant parameters were: inclination angle of the welding wire was 40° and the vertical direction of the laser was 5°. The gap between Cu/SS was set to 0.3 mm to match the wire diameter of 1.2 mm. The offset of the laser beam was set to 0.1 mm on the SS side due to the high reflection of Cu. 

The welding processes for each filler metal were roughly determined in our preparatory experiments. Results of the preliminary experiments showed that welding defects, such as pores, were found in joints under a high welding power. While experiments conducted at a low welding power, resulted in the penetration depth at the root of the weld being insufficient. In this study, in order to connect Cu/SS dissimilar metals, the minimum welding heat input to ensure the formation of a joint for the respective filler material was selected. Parameters used in this study are shown in Table 3. Due to the heat input of joint 1 being larger than joint 2, and welding speed of joint 1 being lower than joint 2, the wire feeding speed of joint 1 was set to 1.5 mm/s and that of joint 2 was set to 1.3 mm/s. 

### 2.3. Characterization of the Welds

After etching with a corrosive liquid (25 g FeCl_3_ + 25 mL HCl + 100 mL H_2_O) for 10–15 s, an optical microscope (OM; ZEISS, Jena, Germany) and a scanning electron microscope (SEM; TESCAN VEGA3, Brno, Czech Rep) with EDX type energy dispersive spectrometer (EDS) were used to observe and analyze the interface formation, microstructure, and fracture morphology of the welded joints. EBSD was conducted to measure the grain size and orientation with the step size of 1 mm. 

Standard ASTM E8/E8m-15a has been applied in the tensile test of the welding joints. In this study, a standard tensile test was performed on an electro hydraulic-servo testing machine (MTS-810, Eden Prairie, MN, USA) and the displacement rate was set at 0.8 mm/min. Corresponding results are shown as the average of five replicated measurements. The dimensions of the test samples are shown in Figure 2. Furthermore, 2D-DIC analysis was performed using the standard DIC test procedures, and the shooting spatial resolution was set to 30 pixels/mm. Images obtained in the 2D-DIC analysis in this study were recorded at 2 Hz. Additionally, the standard ASTM E384-11e1 was adopted to analyze the Vickers hardness of the cross sections of the welded joints. The hardness of the joint in this study was defined under the conditions that the load was 100 g (which equals 1 N) for 15 s, at an interval of 0.05 mm for SS-BM and weld zone, and 0.15 mm for Cu-HAZ and Cu-BM. 

## 3. Results

### 3.1. Macrostructure

The appearances and the cross-sections of the Cu/301 SS joints indicated full penetration in both joints (Figure 3). Our results showed that sound forming of the two joints was achieved. There was a difference in the melting point between the base material and the welding wire; however, in the performed cases, the laser beam was offset on the SS side (Figure 1), and SS-BM and part of welding wire material was melted under the laser heat. The Cu-BM was then melted by the heat conducted from the molten SS-BM and the molten welding wire. Finally, the pure fusion welding mode was generated at both interfaces of the joints. It is noteworthy that the fusion line on both sides of two joints were clearly identified and the joints were free from defects (pores, cracks) (Figure 3), indicating the high adaptability and stability of wire-fed laser welding for both filler metals. Moreover, defect-free joints connected by fusion welding may enhance the mechanical properties of joints. 

### 3.2. Microstructures

Figure 4 shows optical micrographs of the Cu-HAZ. Under the effect of heat conduction, the grains in Cu-HAZ were coarsened to different degrees. Cu-HAZ consists of two areas with different grain sizes. As shown in Figure 4, the lengths of area 1 with a coarse grain size were 609.09 μm and 803.03 μm, respectively, and those of area 2 with a medium grain size were 1318.18 μm and 1951.51 μm, respectively. It is worth noting that the grain size in area 1 at the Cu-HAZ of joint 2 was much smaller compared to joint 1. The grain-coarsening in Cu-HAZ may have influenced the mechanical properties of the joints.

Figure 5 shows optical micrographs of the Cu/weld interface, indicated in Figure 3. Our results showed that obvious fusion lines were found at both C1 and C2 of joint 1 and the columnar grains grew perpendicular to the fusion line (Figure 5a,b). Since the high thermal conductivity of Cu (398 W/mK) would result in a temperature gradient at the Cu/weld, the nucleation and growth of column grains was promoted. Moreover, an unmixed zone (UZ) adjacent to the Cu/weld interface was found (Figure 5a,b) and the coarse grains in the UZ of both the top and bottom parts were integral, indicating that part of the Cu was not melted during the welding process [17]. However, due to the high thermal conductivity of Cu, a large portion of the heat was conducted from the unmelted Cu to Cu-HAZ, and thus led to grain-coarsening in Cu-HAZ from the annealing effect in joint 1. It is worth noting that equiaxed dendrites and columnar dendrites were observed in the SEM analysis (Figure 5c). However, there were different microstructures at the Cu/weld interface (C1, C2) in joint 2. Several SS microstructures were found at the Cu-BM due to the convection effect in the molten pool. Since a portion of heat would be lost through the welding zone, the amount of the heat input at the Cu/weld interface was small and the combined high cooling rate of wire-fed laser welding led to a supercooling effect. There was a partially melted zone (PMZ) composed of melted copper materials and Ni-based phases (Figure 5d,e). It is noteworthy that grain refinement was obtained and is shown in the enlarged micrographs of Figure 5d,e. This may be caused by the heterogeneous nucleation in the promotion of the SS islands. Our results were consistent with a previous study [18]. Moreover, fine spherical steel grains with extremely small sizes were observed in both C1 and C2 (Figure 5d,e) due to the supercooling effect on the Cu/weld interface. Moreover, fine Cu grains and spherical grains were also observed in the SEM micrographs (Figure 5f). The grain refinement would facilitate the hardness elevation at the Cu/weld interface.

Figure 6 shows optical micrographs of the SS/weld interface. The SS/weld interface of both joints were joined by a fusion mode with a typical partially melted zone. As shown in Figure 6a,b, the SS/weld interface exhibited a typical cellular grain growth across the fusion line of joint 1, and, due to the low temperature gradient and high solidification rate of the weld zone, the solidification mode transformed into a mixture of columnar and equiaxed dendrites. As shown in Figure 6d,e, the SS/weld interfaces (S1 and S2) of joint 2 exhibited columnar dendrites and cellular dendrites grew along the maximum temperature gradient from the SS-BM to the weld zone. Significantly, an unmixed zone (UZ) was observed at the SS/weld interface in joint 2 (Figure 6f), and this result was demonstrated by EDS line scanning analysis (Figure 7). As the melting temperature of the weld material was lower than that of SS-BM, a small portion of melted SS-BM metals which was not completely mixed with the weld metals, solidified first; thus, a thin UZ at the SS/weld interface was formed [19]. Compared to the weld metal and BM, an UZ is more susceptible to corrosion [20]. Thus, attempts have been made to eliminate UZs [21]. In this study, the thin UZ at the SS/weld zone interface indicated that a better bimetallic joining was obtained between the SS/weld zone. The solidification modes of the SS/weld interface are totally controlled by solidification parameters, such as temperature gradients and solidification rates.

Figure 8 shows optical micrographs of the weld zones of the joints. Our results showed that columnar dendritic and cellular dendritic solidification modes were formed with distinct solidified sub-grain boundaries in joint 1 (Figure 8a). It is worth recalling that there was a boundary of solidified grains in the middle of the weld, where the microstructures of each side were different. It is obvious that typical cellular grains were on the Cu side while columnar dendrites were on the 301 SS side (Figure 8a), which may have resulted from thermal gradients and supercooling. However, the microstructure in the weld zone of joint 2 was different from that of joint 1 (Figure 8b,c); the weld zone of the SS side was composed of columnar dendrites that grew perpendicularly to the fusion line with primary dendrite arms and secondary dendrite arms (Figure 8b). Additionally, a great quantity of equiaxed dendrites were randomly distributed in the middle of the weld (Figure 8c), which was caused by supercooling due to the dendritic solidification process. Thus, the combined effects received from the constitutional supercooling, as well as the composition and solidification gradients in the fusion zone, produced a complex dendritic solidification structure across the fusion zone.

Figure 9 shows SEM micrographs in the weld zones of joints and Table 4 is the chemical compositions of marked phases in Figure 9. Our SEM analysis showed that the weld of the steel side consisted of a columnar dendritic (Ni, Fe) phase (1A) (Figure 9a) and an equiaxed dendritic (Ni, Fe) phase (1B) (Figure 9b), while the weld of the Cu side consisted of a columnar dendritic (Cu, Ni) phase (1C) and an equiaxed dendritic (Ni, Cu) phase (1D) in the weld zone of joint 1 (Figure 9c). On the SS side of the weld at joint 2, a dendritic (Ni, Fe) phase (1E), and an inter-dendritic phase (1F) of Cu were identified (Figure 9d), while an equiaxed dendritic (Ni, Fe) phase (1G) and a Cu-based matrix (1H) were identified in the middle of the weld (Figure 9e). Moreover, the Cu-based matrix (1J) with a dispersed spherical (Cu, Fe) phase (1I) was identified at the weld on the Cu side (Figure 9f). All the SEM findings are in good agreement with the optical micrographs and they indicate that a perfect bond was obtained between the BMs (Cu and Fe) and the filler material. Moreover, our results showed that there was no solute segregation in the weld of both joints, indicating that a uniform composition of the weld under the convection effect in the welding process was achieved, which was significant to improve the mechanical properties of the joint.

### 3.3. EBSD Microstructural Analysis

#### 3.3.1. Grain Orientations

Figure 10 shows high spatial resolution inverse pole diagrams (IPF) of the local weld sub-regions. The characteristics grains in BM (Cu and SS) and the HAZ obtained by EBSD were consistent with OM and SEM. Our results showed that the Cu-BM was composed of equiaxed copper grains with annealed twins (Figure 10a). Additionally, SS grains in the SS-BM were identified in random orientations (Figure 10b). As shown in Figure 10c,d, grains in the Cu-HAZ of joint 1 exhibited a large size and large aspect ratio, which may have resulted from that the heat conducted through the Cu-BM. Notably, equiaxed grains were observed at the Cu/weld interface, which was consistent with the results in Figure 9c (Figure 10c,d). For the SS/weld interface in joint 1, there was obvious epitaxial crystallization at the bottom (C2) instead of at the top (C1) of the weld (Figure 10e,f), which may have been caused by the supercooling at C2. As shown in Figure 10g, elongated equiaxed grains in the weld zone of joint 1 were observed under the influence of heterogeneous nucleation and the corresponding composition supercooling. As the spatial resolutions of SEM and EBSD were different, this morphology can only be observed in EBSD. As shown in Figure 10h,i, the grain size in Cu-HAZ of joint 2 was much smaller than that of joint 1. It is also worth noting that under the effects of supercooling and heterogeneous nucleation, a number of fine equiaxed grains were generated at both the upper and lower sides of the Cu/weld interface (Figure 10h,i). Since these fine grains existed at the joint, the microhardness at Cu/weld interface would be enhanced. There was obvious epitaxial grain growth perpendicular to the fusion line of the SS/weld interface, and the direction of the growth of the columnar grain was consistent with the direction of heat conduction in joint 2 (Figure 10j,k). As shown in Figure 10l, the heat in the middle of the weld was lost through the weld materials mostly, which resulted in the different temperature gradients at different positions in the weld zone of joint 2. The grains on both sides of the weld showed elongated morphologies while grains in the middle did not present this morphology. This indicated that addition of copper in the welding wire improved the thermal conductivity of the materials in weld zone.

#### 3.3.2. Grain Misorientations

Figure 11 shows the grain boundary distributions of the weld sub-regions. The orientation angles in the range of 2–15° referred to the low-angle boundaries (LABs) of the grain, and the remaining grain boundaries (>15°) were counted as high-angle boundaries (HABs). Our results showed that the proportion of HABs at the SS/weld interface of both joints were high, resulting from the solidification mode of the SS/weld interface being plane solidification and columnar dendritic solidification. The proportion of HABs at the S1 and S2 of joint 1 were 90.1%, 85.4% respectively and that of joint 2 were 86.7%, 80.8% respectively. Thus, formation of the large angle grain boundaries between coarse grains was without the production of sub-grain structures. As the laser beam in the weld was offset to the SS side in both joints, orientation angles in the weld zone of both joints (75.9%, 70.3% of HABs) were similar to those of the SS/weld interface. Significantly, there were twins inside the coarse copper grains, but there was still a relatively high proportion of HABs at the Cu/weld interface (65.4%, 71.2% of HABs) in joint 1. However, the orientation angle at the Cu/weld interface of joint 2 (28.6%, 37.4% of HABs) was different to that of joint 1, which was caused by grain refinement that occurred along the Cu/weld interface under the supercooling effect. The crystallographic relationships between the morphology of grains and corresponding misorientations were highlighted by the local misorientations information.

#### 3.3.3. Grain Size Statistics

Due to the existence of the annealing twins, ranges in the grain sizes of the Cu-BM were wide with an average of 25.62 ± 8.90 µm, and the average grain size of SS-BM was 26.14 ± 6.02 µm. In this study, the average grain size in the weld zone of joint 1 was 42.65 ± 7.87 µm, which was larger than that of joint 2 (36.04 ± 9.26 µm). The average grain size at C1 and C2 of joint 1 were 31.69 ± 17.84 µm and 32.41 ± 16.45 µm, respectively, while the average grain size at C1 and C2 of joint 2 were 22.14 ± 16.20 µm and 21.52 ± 14.06 µm, respectively. Compared to joint 1, the grain size at the Cu/weld interface of joint 2 was much smaller. This was because of the grain refinement in the weld of the Cu side and the grains in the Cu-HAZ of joint 2 was smaller. Different solidification modes, such as equiaxed grains and columnar grains at the Cu/weld interface, resulted in different sizes of grains, which further caused the generation of larger standard deviations of grain sizes at the Cu/weld interfaces of joint 1 and joint 2.

### 3.4. Microhardness

Figure 12 shows the microhardness distribution of the joints. Both joints had their highest hardness values at the SS-BM (nearly about 240 HV), and there was no obvious drop in hardness at the SS-HAZ, which was caused by the high cooling rate of laser welding. The hardness distribution in fusion zone of joint 1 was more uniform than that of joint 2 due to the hardness elevation at the Cu/weld interface of joint 2, which was caused by the grain refinement located at the weld of copper side. The microhardness in Cu-HAZ of both joints was obviously lower than that in the Cu-BM (nearly about 95 HV), which was mainly caused by the annealing effect of the Cu-HAZ under the effect of heat conduction. The grain coarsening in the Cu-HAZ was generated by annealing, which further resulted in a decrease in the dislocation density, thus reducing the microhardness [22]. Therefore, the hardness measured in the local weld zone was related to the grain size formed during the welding thermal cycle. It has been found that a microstructure with a fine grain morphology was more resistant to dislocation movement than that of coarse grains, as boundaries of large-scale grains exist. Hence, fine grains enhanced the hardness at the Cu/weld interface of joint 2. The relationship between the microstructure and the mechanical properties of the weld was indicated by the results of the microhardness and SEM analysis in this study.

### 3.5. Tensile Strength

Figure 13 shows the engineering stress–strain curves (Figure 13a), yield point, tensile strength, breaking strength and Young’s modulus of different joints (Figure 13b). The tensile strengths of joint 1 and joint 2 were 231.3 MPa and 238.7 MPa, respectively, on 88.5% and 91.2% of the copper plate. As shown in Figure 14, there was no obvious deformation in the joint during the tests. Moreover, a larger stretched softened zone was observed in joints filled with ERNiCu-7 filler metal, which indicated that the deformation of the Ni-Cu-based filled joint was higher. As shown in Figure 15, both joints fractured at area 2 of the Cu-HAZ, and the tensile strength of both joints was less than that of Cu due to the softening in the Cu-HAZ, which was caused by grain-coarsening and a decreased dislocation density under heat conduction [23]. As shown in Figure 16, the failure location of both joints exhibited a fibrous surface with dimples, indicating that both joints fractured in a ductile pattern.

Figure 17 shows the progressive 2D-DIC strain maps derived from the joints. Results showed that the strain gradient in the initial joint was very low, and most of the strain was distributed to the Cu-HAZ and the weld zone. However, localization of the strain was obvious, and a large amount of strain accumulated in the Cu-HAZ as the specimen experienced plastic deformation. These results indicated that the Cu-HAZ was weakest zone in the joint. As the 301 stainless steel had a high tensile resistance, there was almost no plastic deformation (strain < 5%) in the SS-BM of joint 1 and joint 2. In addition, due to the high tensile resistance of the Ni-based solid solution in the weld of joint 1, the metal strained within 6% and it is interesting to note that the strain concentration of the Cu/weld interface was more obvious in joint 1 than that in joint 2.

Previous studies [24] showed that the softening of the material intensified with the increase in heat input, thus raising the strain of the material. However, in this study we found that the welding heat input of joint 2 with the higher plastic deformation was lower than that of joint 1 (Figure 13). Therefore, there must be other reasons for the high plastic deformation of joint 2. It can be concluded from the 2D-DIC analysis that two possible reasons may result in the enhancement of the plastic deformation in joint 2. First of all, the overall deformation was enhanced by the plastic deformation of the weld zone in joint 2. On the other hand, the softer weld zone of joint 2 may relieve the severe residual stress at the Cu/weld interface. During the tensile process, the interface between the stiff material and the soft material generated the stress concentration [25], and severe stress concentration at the Cu/weld interface of joint 1 was detected during the tensile process (Figure 17). The strain was extremely uneven. Since the Cu-based matrix in the weld zone of joint 2 was identified, plastic deformation was more uniform in joint 2, easing the residual stress at the Cu/weld interface, and finally improved the plastic deformability of the joint. In conclusion, since the weld zone of joint 2 exhibited great plastic deformation, the coordinated deformation of the weld zone could relieve the stress concentration generated at joint 1 to some extent. Thus, the joint could undergo great plastic deformation before fracture.

## 4. Conclusions

The results showed that there were no obvious weld defects in the fusion zone and the welding interface of the wire-fed laser welded joint with 600 alloy filler metal and ERNiCu 7 filler metal. All the optical and SEM micrographs agree with these findings. In conclusion, 600 alloy filler and ERNiCu-7 filler were appropriate filler materials for the welding of copper to stainless steel.Cu-HAZ consists of two areas with different grain sizes and the sizes of grains in Cu-HAZ of the Ni-Cu-based filled joint was smaller than that of the Ni-based filled joint. Our results showed that the composition distribution in the weld zone of both joints was uniform and the weld zone was composed of a mixture of columnar and equiaxed dendrites. Epitaxial grain growth was identified along the SS/weld interface.EBSD analysis of SS/weld interfaces and the fusion zone indicated that all these regions have a distinct grain morphology and the grain refinement zone was spotted in the Cu/weld interface of the Ni-Cu-based filled joint.Results of the microhardness measurements indicated that the highest hardness was observed at SS-BM (nearly 240 HV), among all the weld zones. Additionally, the hardness in the weld zone of the Ni-based filled joint was more homogeneous than that of the Ni-Cu-based filled joint, and there was an elevation in hardness at the Cu/weld interface of the Ni-Cu-based filled joint due to the grain refinements located in the weld of Cu.A 91.2% tensile strength of Cu was obtained in the Ni-Cu base filled joint. Among these joints, most of the strain was distributed to the Cu-HAZ and fractured in a ductile pattern. Analysis of the 2D-DIC showed that the deformation of the Ni-Cu-based filled joint was higher than that of the Ni-based filled joint, which was caused by the high deformation in the weld zone and less stress concentration at Cu/weld interface.

## Figures and Tables

**Figure 1 materials-14-02122-f001:**
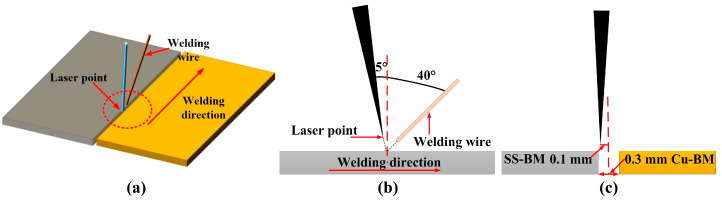
Schematic overview beam offset application.

**Figure 2 materials-14-02122-f002:**
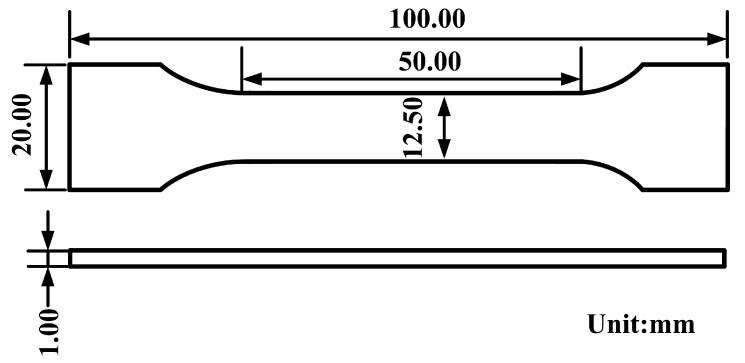
Specimen dimension for tensile test.

**Figure 3 materials-14-02122-f003:**
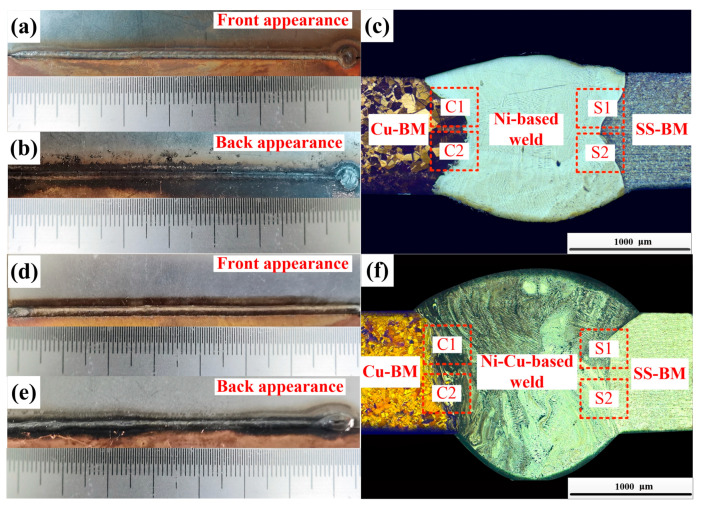
The appearances and the cross-section of the joints.

**Figure 4 materials-14-02122-f004:**
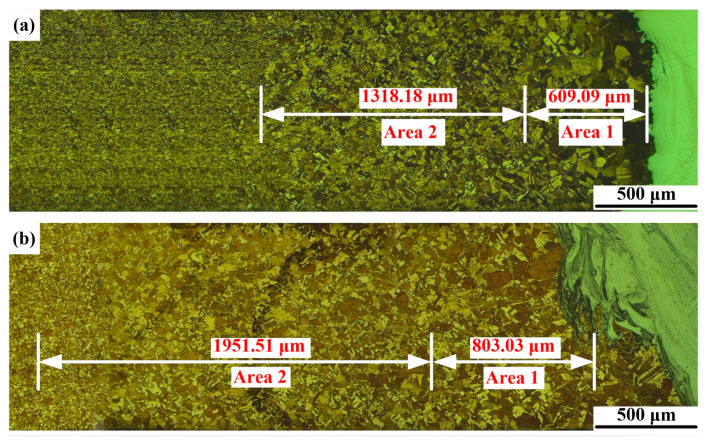
Optical micrographs of the Cu-HAZ: (**a**) joint 1 and (**b**) joint 2.

**Figure 5 materials-14-02122-f005:**
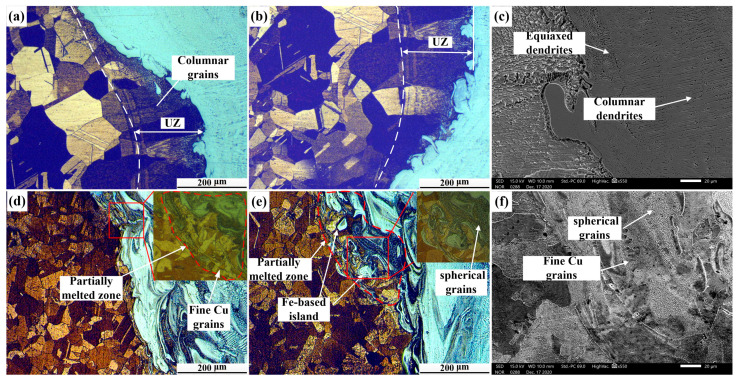
Optical micrographs of the Cu/weld interface in joints: (**a**) optical image of C1 in joint 1; (**b**) optical image of C2 in joint 1; (**c**) SEM image of joint 1; (**d**) optical image of C1 in joint 2; (**e**) optical image of C2 in joint 2; (**f**) SEM image of joint 2.

**Figure 6 materials-14-02122-f006:**
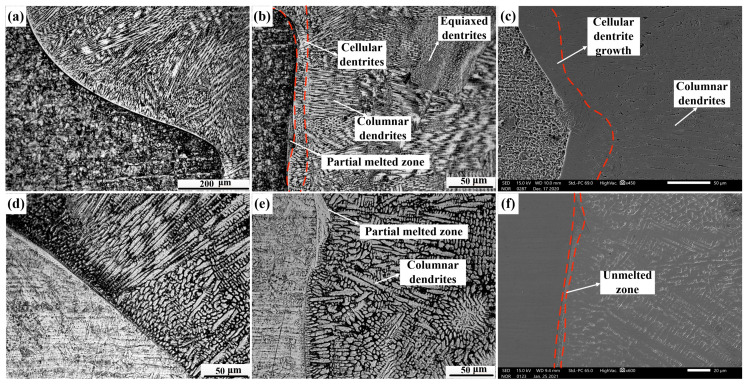
Optical micrographs of the SS/weld interface in joints: (**a**) optical image of S1 in joint 1; (**b**) optical image of S2 in joint 1; (**c**) SEM image of joint 1; (**d**) optical image of S1 in joint 2; (**e**) optical image of S2 in joint 2; (**f**) SEM image of joint 2.

**Figure 7 materials-14-02122-f007:**
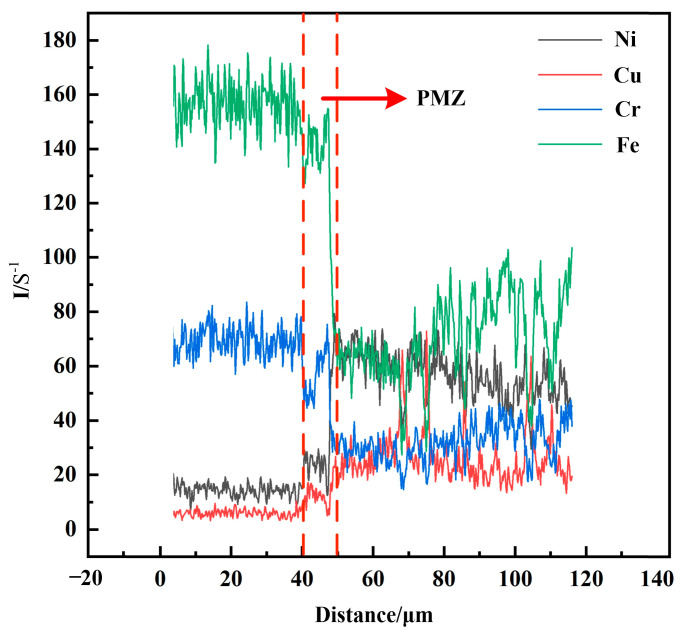
EDS cure of SS/weld interface in joint 2.

**Figure 8 materials-14-02122-f008:**
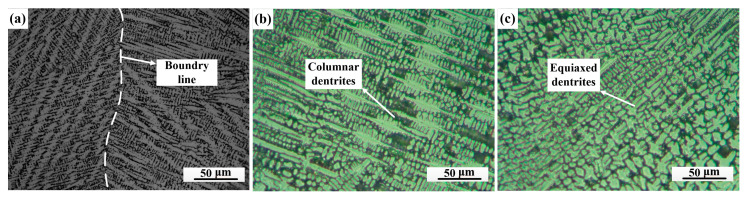
Optical micrographs of weld zones: (**a**) joint 1 and (**b**,**c**) joint 2.

**Figure 9 materials-14-02122-f009:**
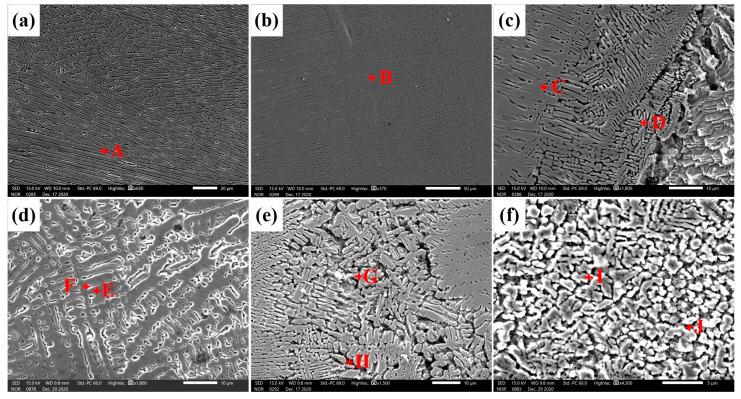
SEM micrographs in the weld zones: (**a**–**c**) joint 1 and (**d**–**f**) joint 2.

**Figure 10 materials-14-02122-f010:**
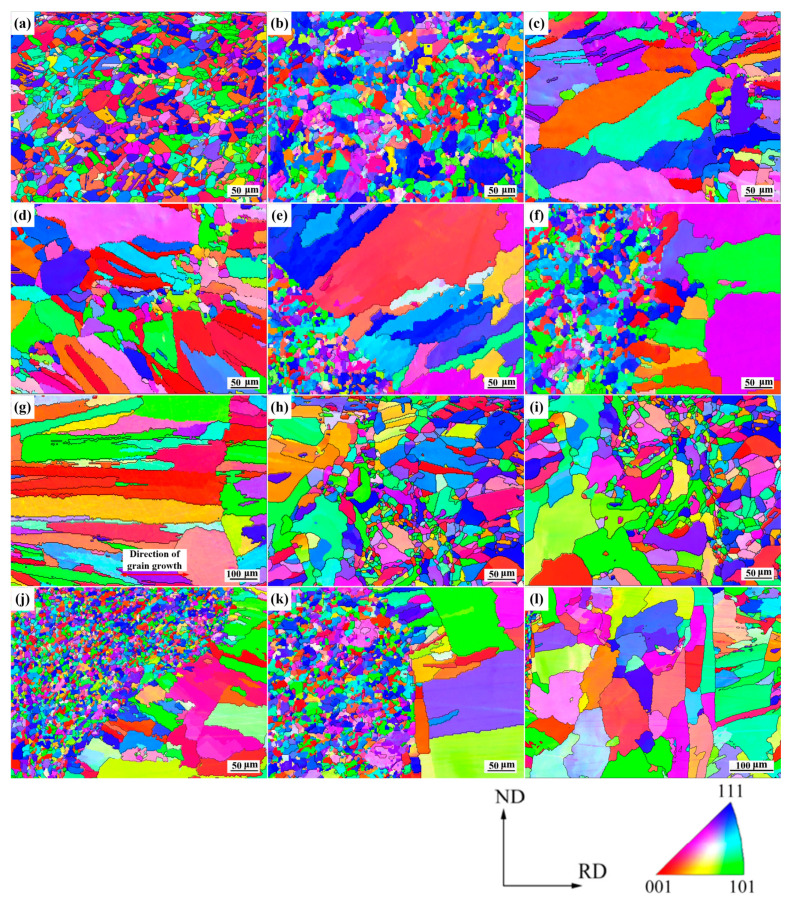
EBSD-IPF micrographs of local weld sub-regions: (**a**) Cu-BM; (**b**) SS-BM; joint 1: (**c**) C1; (**d**) C2; (**e**) S1; (**f**) S2; (**g**) weld zone; joint 2: (**h**) C1; (**i**) C2; (**j**) S1; (**k**) S2; and (**l**) weld zone.

**Figure 11 materials-14-02122-f011:**
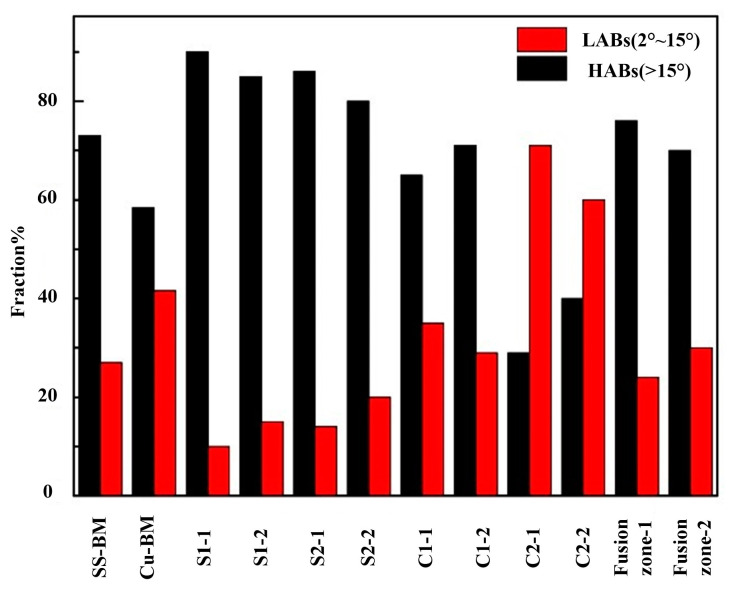
Grain misorientation distributions of local weld zones.

**Figure 12 materials-14-02122-f012:**
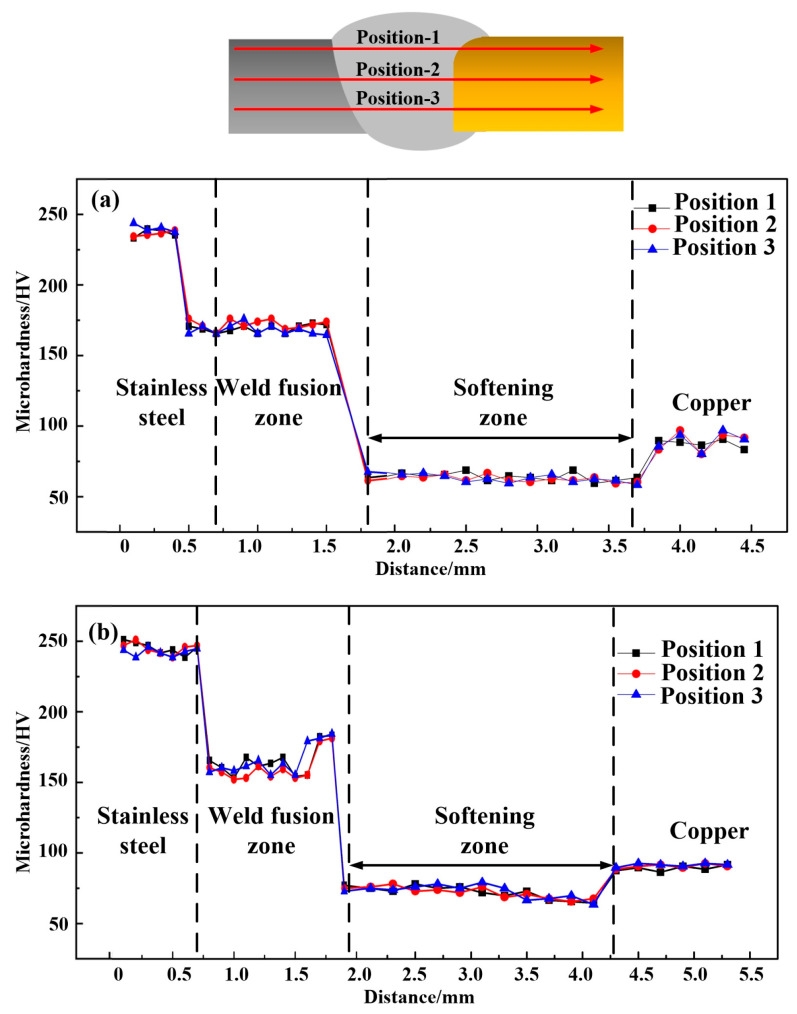
Microhardness distributions across the welds of joints: (**a**) joint 1 and (**b**) joint 2.

**Figure 13 materials-14-02122-f013:**
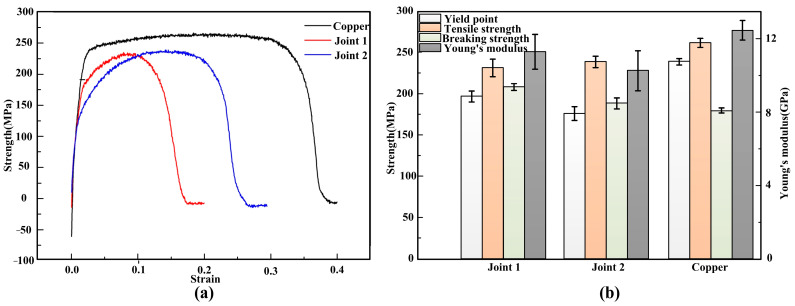
(**a**) The engineering stress–strain curves and (**b**) yield point, tensile strength, breaking strength and Young’s modulus of different joints.

**Figure 14 materials-14-02122-f014:**
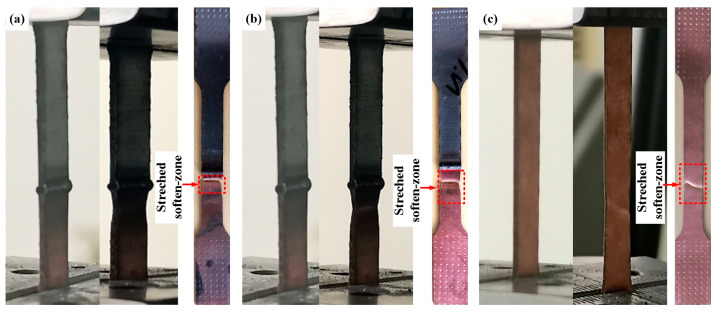
The specimens of the joints mounted on the testing machine at the beginning of the test, during the test and at the end: (**a**) specimen of 600 alloy filled joint; (**b**) specimen of ERNiCu-7 filled joint; and (**c**) specimen of pure Cu.

**Figure 15 materials-14-02122-f015:**
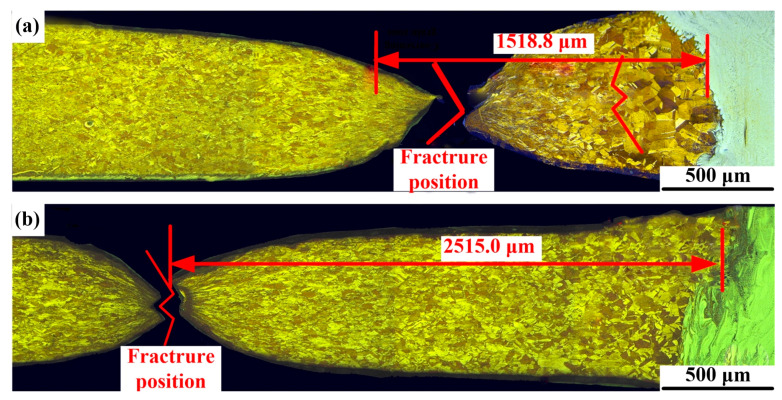
Fracture position of joints: (**a**) joint 1 and (**b**) joint 2.

**Figure 16 materials-14-02122-f016:**
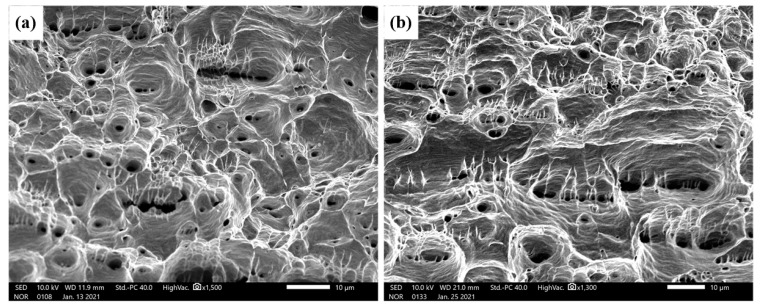
SEM micrographs of the fracture surfaces of the joints: (**a**) joint 1 and (**b**) joint 2.

**Figure 17 materials-14-02122-f017:**
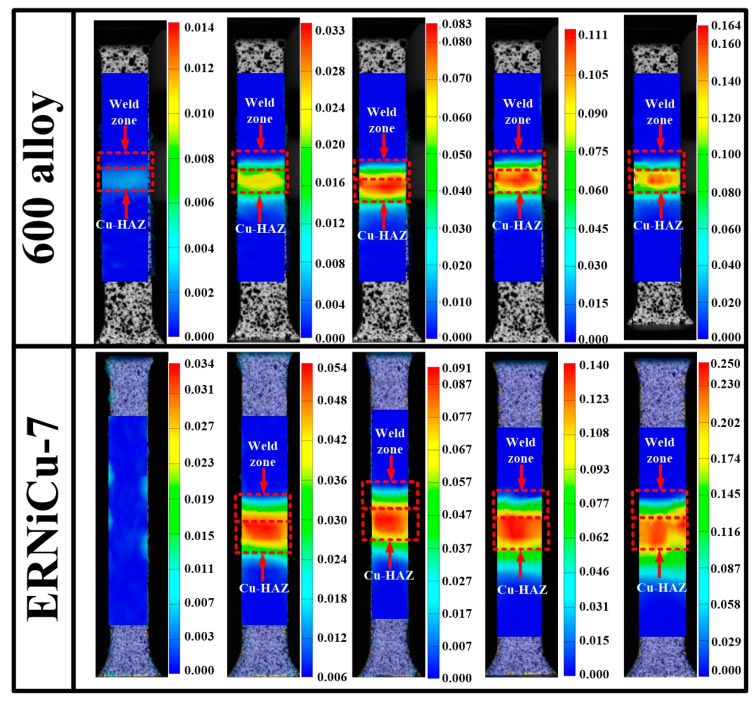
Sequential strain maps obtained through 2D-DIC.

**Table 1 materials-14-02122-t001:** Chemical compositions of the base metals (wt.%).

**Name**	**C**	**Mn**	**Si**	**Cr**	**Ni**	**P**	**S**	**Fe**
301SS	0.053	1.300	0.703	18.000	8.220	0.033	0.002	Bal
**Name**	**O**	**S**	**Zn**	**Fe**	**As**	**Sb**	**Bi**	**Cu**
T2 copper	0.060	0.005	0.005	0.005	0.002	0.002	0.001	Bal

**Table 2 materials-14-02122-t002:** Chemical compositions of the filler metals (wt.%).

**Name**	**Ni**	**Fe**	**Cu**	**Cr**	**Mn**	**Nb**
600 alloy	67	3	-	20	3	3
**Name**	**Ni**	**Fe**	**Cu**	**Si**	**Mn**	**P**
ERNiCu-7	69	2.5	23.215	1.25	4	0.02

**Table 3 materials-14-02122-t003:** Welding parameters used in this experiment.

Filler Metal	Samples	Welding Power (W)	Welding Speed (m/s)	Wire Feed Speed (mm/min)	Heat Input (KJ/m)
600 Alloy	Joint 1	1400	0.018	1.5	77.78
ERNiCu-7	Joint 2	1500	0.020	1.3	75.00

**Table 4 materials-14-02122-t004:** Chemical compositions of marked phases in Figure 9 (at.%).

	Fe	Ni	Cu	Cr
A	31.50	38.27	15.10	14.26
B	29.01	34.57	16.10	13.61
C	20.00	31.62	22.68	10.48
D	18.16	28.54	31.68	11.42
E	35.66	27.88	20.29	14.87
F	16.71	16.61	61.81	10.25
G	28.03	31.33	16.66	12.87
H	12.12	10.26	65.92	11.7
I	21.66	32.02	34.18	11.05
J	15.11	15.72	68.86	10.6

## Data Availability

Data are contained within the article.

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
