# Peer review of "Wire-Feeding Laser Welding of Copper/Stainless Steel Using Different Filler Metals"

_materials, 2021, doi:10.3390/ma14092122_

Round 1
Reviewer 1 Report
This paper presents research on laser welding of copper and 301 stainless steel using two kinds of filler metals: Ni-based and Ni-Cu-based. The mechanical properties of the joints were determined by means of the standard tensile test, and also the Vickers hardness test was carried out. Moreover, during the tensile tests, the standard digital image correlation (DIC) test was involved. The interface formation, microstructure, and fracture morphology were observed by means of a scanning electron microscope with EDX. Moreover, the EBSD characterization was carried out. It was found that different grain growth depended on kind of the filler metal. The joint made using the Ni-Cu-based filler shown higher strength properties and exhibited higher plastic deformation. Moreover, that was observed that all fractures of the joints were ductile type. The manuscript is written very well, so in my opinion, this manuscript can be printed when the Authors take into account the following comments:
• In the article is a lack of the information which parameters were used for which filler metal
• Table 1 and 2: in the third line “Name, etc….” should be bolded
• Figure 3: the visibility of the designations has to be improved
• Figure 4: the visibility of the designations has to be improved; overall quality of the micrographs should be improved; “cu” has to be changed to “Cu”; the regions of the equiaxed and columnar dendrites in the picture (c) are invisible, probably because of poor quality of the picture
• Figure 7: the visibility of the designations has to be improved
• Figure 10 should be moved after the chapter where it was mentioned
• On page 11 “Hv” has to be changed to “HV”, both in the text and in Figure 11
• In line 363 on page 13 the included information concern Figure 13, not 12
• Figure 15: the visibility of the scale has to be improved.
Reviewer 2 Report
I am asking the authors to add nomenclature to the paper.
A very good introduction with a decently edited literature review.
The heading of table two is not on the correct page - line 88 of the manuscript.
The presented description of the macrostructure does not raise any objections, however the quality of the figures could be different - figure 3 - please delete the edges from the table, pay attention to the position of the caption - it is not on the page it should be - line 141 of the manuscript. Here you can work on the quality of the added indications. I address similar remarks to Figures 4-7 - I have no objection to the description of the microstructure itself and the analysis carried out by the authors.
In Figures 2, 9, 10, 11 and 12, authors should standardize the size, type and formatting of the font. The authors should also work with the added descriptions in Figures 13 and 14 - they are illegible, some markings are too small - maybe they should be placed differently in the paper, even at the expense of increasing the volume of the manuscript.
Figure 15 is also not very legible - especially the scales related to deformations. This drawing needs to be corrected.
I have comments on paragraph 3.5. I believe it is too poor. The authors should show in the manuscript a photo of the specimen mounted on the testing machine - at the beginning of the test, during the test and at the end, so that the reader can assess the behavior of this specimen - see if it is deformed during the test itself. To compare certain test results, authors should make at least 3-5 copper specimens and at least 5 welded specimens, preferably 8 or more, to evaluate scattering of results. They should determine the Young's modulus, yield point, tensile strength and breaking stress - I presume that the specimens broke. In Figure 12, I miss a photo of a specimen made of copper. I believe that it should be an additional figure with such photos. The authors should break a total of at least 13 specimens, determine typical constants determined during the tensile test, statistically evaluate the results obtained, and provide the scatter of the results. A full error account should be made here. It is worth completing this - but I don't think the authors are able to fulfill my request. Giving data for only three different specimens is a misunderstanding at this point - especially drawing conclusions based on these results, especially for welded structures.
Nevertheless, I believe that the paper can be submitted for publication, but after making small corrections, as I wrote above. I suggest minor revision.
Reviewer 3 Report
Many shortcomings have been noticed, as follows:
- It is not appropriately explained the originality level of the investigation and the knowledge progress in the welding field. There are many articles focused on welding/brazing Cu to Steel.
- Given the difference between the melting temperatures of the base materials, it is more likely that the joint between fusion metal/weld and stainless stell to be achieved by brazing, not welding, especially when the filler material is a Cu-Alloy. New information and explanations are needed.
- Because of the small thickness of sheets, which is 1 mm, it is unclear why the authors decided to use filler metals for joining thin parent materials.
- It is strongly recommended to indicate the standards used for describing the chemical composition. The materials have to be specified, according to the standards employed in selecting the materials.
- The pulse frequency and the heat input are not provided in the manuscript. Also, the laser beam offset is not specified.
- Table 3: The “Feed” has to be replaced with “Wire feed speed”. It is not explained the wire feed speed decreases when the laser power increases. It is unclear which filler materials have been used to achieve the Joint 1 and the Joint 2.
- Figure 11: The sketch of welded joints is inappropriately located on figure and that creates a misunderstanding in terms of indentation test arrangement in the joints’ regions.
- Figure 13: The tensile testing revealed that the fracture was located in the Copper material unaffected by the welding process. So, it is not clear why the tensile testing results were 88.5% and 91.2%, comparing to the Copper base material strength if the fracture was located in the parent material. On the other hand, a contradictory statement has been noticed “This proves that the Cu-HAZ was the weakest zone in the joint” that seems to be a contradiction between explanations.
- Many technical errors have been found. For instance, the stainless steel is resistant to corrosion, not to erosion which is a term specific to soil, stones etc (L29).
- The translation into English language has to be revised by a native speaker.
Reviewer 4 Report
My comments for the manuscript 1178625 are as follows.
- Page 2, Line84: (Ni-based) and (Ni-Cu-based) should be change to (Ni-based filler) and (Ni-Cu-based filler).
- Table 1.: Chemical compositions of 301 stainless steel need to be specify the main element of Fe balance. (Fe bal.)
- Figure 1.: Please specify (write) the angles of welding wire of 40°and laser vertical direction of 5°in Figure 1.
- Table 3.: Authors need to specify that Joint 1 is welded by Ni-based filler and Joint 2 is welded by Ni-Cu-based filler in Table 3. In other words, please add the column of filler metal in Table 3.
- Figure 3.: I cannot easily understand the materials combinations of two joint. Please, add “Cu-BM/ Ni based weld/ SS-BM” and “Cu-BM/Ni-Cu based weld/SS-BM” corresponding to each photo of welding joints in Figure 3.
- Figure 4.: The six photographs must be corresponded to C1 and C2 of Figure 3. The right materials are weld metal and left materials are Cu base metal in Figure 3, but in Figure.4 all photographs are reversed. Please revise them.
- Page 8, Line230-231: In the text, there is “As shown in Figure 8 (a), the Cu-BM was composed of equiaxed copper grains with annealed twins. And the SS grains in the SS-BM had many random orientations (Figure 8 (b)).” However, the caption of Figure 8 says “(a) SS-BM; (b) Cu-BM;”. Please check the mismatch.
- Page 8, Line 239-241: Authors explained “As shown in Figure 8 (g), there were elongated equiaxed grains”. Please mention the grain grows direction of the elongated grains if it possible.
- Figure 11: Please explain in the text about the reason why there is a difference between two hardness of copper base metal of joint 1 and joint 2 outside of HAZ.
- Page 11, Line 325-328: Please explain why both joint didn’t fracture at softening area of course grain in Figure 11.
- Figure 15: I confuse the strain map, because the vertical positions of strain at left color map and right tensile test pieces. Please coincide vertically both strain positions of color maps and tensile test price.
Round 2
Reviewer 3 Report
Based on the reviewers' comments, the content of the manuscript has been improved. In my opinion, this updated variant can be considered to be published in the Materials Journal.